# Challenges in the Isolation and Proteomic Analysis of Cancer Exosomes—Implications for Translational Research

**DOI:** 10.3390/proteomes7020022

**Published:** 2019-05-15

**Authors:** Jadwiga Jablonska, Monika Pietrowska, Sonja Ludwig, Stephan Lang, Basant Kumar Thakur

**Affiliations:** 1Translational Oncology, Department of Otorhinolaryngology, University Hospital Essen, 45147 Essen, Germany; ludwigsonja@gmx.net; 2Center for Translational Research and Molecular Biology of Cancer, Maria Sklodowska-Curie Institute–Oncology Center, Gliwice Branch, 44-100 Gliwice, Poland; Monika.Pietrowska@io.gliwice.pl; 3Department of Otorhinolaryngology, Head and Neck Surgery, University Hospital Essen, 45147 Essen, Germany; stephan.lang@uk-essen.de; 4Cancer Exosome Research Lab, Department of Pediatric Hematology and Oncology, University Hospital Essen, 45147 Essen, Germany

**Keywords:** proteomics of exosomes, heterogeneity of exosomes, extracellular vesicles, cancer, tumor progression

## Abstract

Exosomes belong to the group of extracellular vesicles (EVs) that derive from various cell populations and mediate intercellular communication in health and disease. Like hormones or cytokines, exosomes released by cells can play a potent role in the communication between the cell of origin and distant cells in the body to maintain homeostatic or pathological processes, including tumorigenesis. The nucleic acids, and lipid and protein cargo present in the exosomes are involved in a myriad of carcinogenic processes, including cell proliferation, tumor angiogenesis, immunomodulation, and metastasis formation. The ability of exosomal proteins to mediate direct functions by interaction with other cells qualifies them as tumor-specific biomarkers and targeted therapeutic approaches. However, the heterogeneity of plasma-derived exosomes consistent of (a) exosomes derived from all kinds of body cells, including cancer cells and (b) contamination of exosome preparation with other extracellular vesicles, such as apoptotic bodies, makes it challenging to obtain solid proteomics data for downstream clinical application. In this manuscript, we review these challenges beginning with the choice of different isolation methods, through the evaluation of obtained exosomes and limitations in the process of proteome analysis of cancer-derived exosomes to identify novel protein targets with functional impact in the context of translational oncology.

## 1. Introduction

Emerging evidence shows that exosomes are powerful key players regulating multiple processes during carcinogenesis and tumor spread [1,2]. Protein cargo of exosomes play an important role in all of these processes [3]. Surface proteins of exosomes target specific adjacent and distant sites, such as metastasis. Intraexosomal protein cargo is considered to facilitate the crosstalk between tumor and environment, e.g., cells present in the premetastatic niche [4]. Uptake of exosomes by the cells lead to the release of exosomal cargo inside the cells and in turn to the activation of the downstream signaling pathways relevant in tumorigenesis and metastasis [1,5]. Therefore, a detailed characterization of protein cargo on the homogeneous pure population of exosomes is essential in order to understand correctly the mechanisms behind communication between cancer and immune cells. Herein, we will discuss the limitations and challenges in performing proteomics of exosomes, which starts with the choice of sample for exosome isolation and isolation method, and further depends on the protein characterization and enrichment criteria. The quality of the proteomics data can then be useful for basic and clinical research in order to characterize and classify exosomes based on the proteomic profiles. This will allow for further evaluation of exosomes in diagnostics as biomarkers or therapeutics.

## 2. Exosomes and Their Function

Exosomes are small non-plasma membrane-derived vesicles (30–120 nm) of endocytic origin [6]. Production of exosomes is a very tightly regulated process governed by multiple signaling molecules and occurs at the side of early endosome, resulting in the formation of a multivesicular body (MVB). After the fusion of MVBs with the plasma membrane, intraluminal vesicles (ILVs) from MVBs are released [7]. Two major pathways are suggested in the production of exosomes at the endosomal membrane: the endosomal sorting complex required for transport (ESCRT)-dependent pathway and the ESCRT-independent pathway [8]. Exosomes were discovered first in 1983 by two independent groups reporting that maturing blood reticulocytes release transferring receptors associated vesicles (exosomes) in extracellular space [9,10]. Since these findings, the field of exosome research has exploded and in the last three decades, thousands of publications related to understanding exosome biology in health and disease have appeared. With the recent advancement in the isolation techniques and characterization tools, it appears that a major set of publications deal with a heterogeneous mixture of vesicles containing non-exosome membrane vesicles and exosomes.

Exosomes carry different classes of molecules, including proteins, metabolites, and nucleic acids [4,11]. There are discussions about whether the composition of exosomal cargo only reflect the phenotype of parental cells or are actively regulated to fulfill different regulatory and communication functions [12]. Nevertheless, the cargo of exosomes reflects, to some extent, that of the parental cells, which allows one to determine their origin. For this reason, tumor-derived exosomes (TEX) are an important non-invasive surrogate marker for tumors enabling them to serve as a sort of liquid biopsy [4]. Most interestingly, exosomes are involved in many aspects of intercellular communication as they may transmit a complex network of signals driving cell death, survival, and differentiation between a secreting cell and multiple types of neighboring or distant recipient cells. It is noteworthy that exosomes are known to be involved in the regulation of the immune response by affecting multiple immune cells, such as dendritic cells, T cells, B cells, NK cells, and macrophages [2].

Due to the recent improvement in EV isolation and analysis techniques, several key papers have come up with findings that would change the fields view on exosomes as a simple mixture of EVs. Zhang, H. et al. reported that exosome isolation using classical ultracentrifugation method, when subjected to the asymmetric flow field flow fractionation, could be divided into three different subpopulations—two exosome populations: large (Exo-L, 40–120 nm) and small (Exo-S, 60–80 nm), and “exomeres” that are non-membrane nanoparticles (~35nm) [13]. In another report, detailed analysis of various EVs and non-membranous particles isolated to homogeneity revealed a requirement for the reassessment of exosomal protein, RNA, and DNA composition that was previously associated with exosomes [14]. Both of these studies, besides transcriptomics and genomics, clearly highlight the importance of proteomic approaches to define different populations of vesicles and nanoparticles, which are released in the extracellular space.

## 3. Source and Methods of Isolation for Proteomic Analysis

For every exosome-based study, several parameters, such as source, methods of isolation and downstream analysis, have to be considered carefully. The choice of one parameter or another can lead to an entirely different outcome in the planned experiment. Exosomes can derive from either cell/tissue culture supernatants or primary body fluids, like blood or urine. The use of well-characterized cell lines is a routine procedure in most laboratories working in cancer research areas as an easy and safe method to start experiments, and to acquire reproducible initial data. Analyses of tumor-derived exosomes that require large quantities of plasma exosomes of patients could also be initially based on cell lines. In the field of exosome-based target identification, proteins cannot be amplified such as other molecular counterparts, e.g., DNA and RNA in PCR. Therefore, the evaluation of the proteomic content of exosomes proves to be a challenging task.

Nevertheless, the major disadvantage of cell lines is that they do not reflect the complexity of “in vivo” systems, consisting of artificially created homogenous immortal cells and frequently lacking important molecules that could be identified as biomarkers. Therefore, functional analyses based entirely on the cell line-derived exosomes can be misleading and require re-evaluation in patient material. This shows a strong need to standardize protocols used for exosome isolation and analysis from biologically relevant bodily fluids, such as blood [15], urine [16], breast milk [17], cerebrospinal fluid (CSF) [18], saliva [19], human semen [20], and synovial fluid [21]. All these different sources of exosomes require meticulous analysis, which will result in the decision of the best criteria for the exosome isolation method as well as further exosome analysis.

Conditioned media obtained from cell culture often contains fetal bovine serum (FBS), which is enriched in bovine serum exosomes. These exosomes can lead to misleading target detection by high sensitive mass spectrometers. Nevertheless, most of the established cell lines, irrespective of their origin, require FBS. This problem is hard to solve, as growing under serum starvation would be one option to avoid bovine-derived exosomes in proteomic analysis, however, it is known to activate a plethora of signaling pathways related to stress response. This intracellular stress generated due to lack of serum can completely alter the metabolic behavior of the cell and in turn influence exosome secretion pathways, leading to misrepresentative experimental outcomes. Therefore, culturing under serum-deprived conditions can only work for some cell lines that are specifically generated for such growth conditions. An alternative would be to deplete bovine exosomes from FBS, which is currently done by the ultracentrifugation of FBS at high speed (100,000× to 120,000× *g* for 2 to 18 h), depending on the experimental requirement. Shelke et al. have shown that for transcriptome analysis of exosomal RNA, ultracentrifugation at 120,000× *g* for 18 h led to almost 95% decrease in the signal background from FBS EV [22].

Blood plasma and serum constitute the most promising source of exosomes, as they include circulating exosomes originating from blood cancers as well as solid tumors [23]. Blood can be easily obtained from patients, except in children when a high blood volume is required. However, if no proper EV isolation technique is applied, plasma and serum-derived EVs contain various types of chylomicrons, lipoprotein particles, and plasma proteins that can dramatically reduce the sensitivity of mass spectrophotometry [24,25]. Moreover, as blood contains proteases, nucleases, and lipases, the method to draw blood and subsequently to handle the sample should be optimized for proteomics.

Urine is considered as another promising body fluid that could be used in the future as an alternative source of exosomes. Obtaining a urine sample from patients is a simple and non-invasive process. Urine is clean and is generally free of proteases, nucleases, and lipases, which makes the storage and processing of proteins much easier than blood. However, the presence of protein uromodulin in the exosome preparations from urine poses a major problem in downstream proteomics analysis, as a high uromodulin level enriches the mass spec signal and thereby limits the effectiveness of identification of exosome-associated protein targets [17,26]. Recently, thermochemical and centrifugal based methods have been applied to deplete uromodulin protein from urine, but these methods are labor- and time-consuming and lead to compromises in exosome quality [26,27]. Another approach, which can be applied to eliminate contaminating proteins from body fluid is m/z exclusion limit [28]. Hiemstra et al. have recently applied this approach to set up m/z exclusion list (ExL) of uromodulin-related peptide ions, which effectively improved the specificity of the mass spectrometry to detect exosome-associated proteins in urine without the requirement of physical elimination of the uromodulin contamination [28].

Cerebrospinal fluid (CSF) [18] can be another useful source of exosomes, especially in neurological cancers and disorders [29,30]. Because exosomes from primary central nervous system (CNS) cancers can directly infiltrate the CSF, a large amount of attention was paid recently to CSF-derived exosomes as a diagnostic and prognostic biomarker in CNS cancer. Due to the invasive and painful procedure of drawing CSF samples from patients and the requirement for multiple subsequent draws to obtain sufficient material, it is challenging, especially in pediatric patients. Similar to blood and urine, the presence of high immunoglobulin concentrations can greatly interfere with mass spectrometry analysis of the CSF-derived exosomes [30]. Ultracentrifugation and later incubation of CSF-derived exosomes from protein G agarose beads were shown to remove the majority of immunoglobulins from exosome samples, and thereby to increase the signal of exosome-specific peptides in mass spectrometry analysis [30].

Realizing that ultracentrifugation, which used to be the gold standard for exosome isolation, provides a mixture of heterogeneous membrane-bound exosome-like EVs with contaminants in the preparation, a large number of protocols for exosomes purification have been proposed in the last decades. Each method of exosome isolation is based on the criteria of improving yield and scalability of the exosome preparation. Isolation methods that evolved after the classical ultracentrifugation (UC) [31] are: density-gradient centrifugation (DGC) [32], sucrose cushion centrifugation [31], size exclusion chromatography (SEC) [33,34], affinity chromatography (AC) [35,36], membrane filtration [37] and recently established AF4 technique [13,38]. From all of these methods, it is suggested that SEC is one of the most efficient methods to retain high concentrations of biologically-active exosomes from plasma specimens [39]. For a comprehensive summary of exosome purification protocols based on exosome yield, purity, and scalability, please refer to a review by Xu et al. [38]. Although the choice of exosome isolation protocol exclusively depends on the downstream requirement and the source of the exosomes, the initial decision to choose a given method is decisive for the outcome of the whole experiment. AC that uses beads conjugated to the exosome-specific markers could have an advantage over other methods, which are in principle based on non-specific physical properties of exosomes [35,36]. Hence, once the cancer-specific surface marker on the exosome is established, one could include specific antibodies on the beads, in addition to the exosome’s specific markers, in order to obtain cancer-released exosomes. Although AC has an advantage over other methods, in this case, specific surface markers have to be defined for each cancer model exosome, to distinguish cancer specific exosomes from other EVs (microvesicles or apoptotic bodies), or exosomes from healthy cells. At present, there is no existing method, which can yield pure cancer-specific exosome material for downstream proteomic analysis. To overcome this issue, several groups have come up with an alternative approach to combine two or more methods to achieve high exosome purity [40,41,42]. Of note, proteomic profiling of EVs purified by a combination of ultrafiltration and size exclusion chromatography revealed high EV purity in comparison to EVs purified by classical ultracentrifugation method [40].

After isolation of exosomes from the biological sample of choice, the quantity and purity of the exosomes should be evaluated. Exosomes can be quantified by measuring protein concentrations in isolated exosome fractions using a BCA protein assay kit, nano tracking analysis (NTA), dynamic light scattering (DLS), flow cytometry or transmission electron microscopy [43]. Nevertheless, none of the biochemical and microscopy-based approaches to quantify exosomes can provide information on the amount of pure exosomes in the mixture of EVs. Only in combination with exosome specific fluorophore antibodies in light microscopy or immunogold staining in electron microscopy can one obtain precise information about the quantity of exosomes in the EV mixture and thereby the purity of the exosome preparation [44]. In addition, recent research highlights the importance of imaging flow cytometry for the analysis of homogenous EVs in the samples containing heterogeneous EVs and non-EV particles [45,46].

## 4. Proteomics of Exosomes

In the past decades, proteomic analysis is emerging as a powerful tool to explore and analyze protein targets playing a functional role in triggering signaling pathways regulating health and disease status of a cell [47]. For example, mass spectrometry-based proteomic analyses of cancer tissue have led to the establishment of several cancer-specific biomarkers that can be used for detection or therapy of cancer at early or late stages of disease [48,49,50]. The invasive approach to obtain cancer tissue for proteomic analysis is painful for patients; therefore, liquid biopsy from body fluids (a rather non-invasive approach) could serve as an important alternative to identify cancer biomarkers. In agreement with this, it was shown that the amount and content of exosomes isolated from plasma/serum [51], urine [52], or ascites fluids [53] of cancer patients positively correlate with the tumor progression. According to the available studies, exosomes contain numerous proteins that are involved in intracellular membrane fusion and transport, such as flotillins, annexins, GTPases, Rab, and SNAREs; heat shock proteins (Hsp60, Hsp70, and Hsp90) and proteins involved in multivesicular body (MVB) biogenesis, such as ALIX or Tsg101 [54]. Moreover, membrane-microdomain associated proteins, particularly tetraspanins (CD9, CD63, CD81, and CD82) are present [54]. Importantly, a systematized database of exosome proteins, RNAs and lipids proteins from different sources was created—ExoCarta [http://www.exocarta.org].

Analysis of exosomes from a wide variety of cells and body fluids have allowed the identification of several classes of proteins: (1) membrane adhesion (e.g., integrins); (2) membrane trafficking (e.g., annexins, Rab protein family); (3) cytoskeletal components (e.g., actins, ERM proteins); (4) lysosomal markers (e.g., CD63, LAMP-1/2); (5) antigen presentation factors (e.g., HLA class I and II/peptide complexes); (6) tumor antigens (e.g., MelanA/Mart-1, gp100, CEA, HER2); (7) death receptors (e.g., FasL, TRAIL); cytokines and cognate receptors (e.g., TNFα, TNFR1, TGF-β); iron transport factors (e.g., TfR); (8) enzymes (e.g., pyruvate kinase, enolase); (9) heat shock proteins; (10) drug transporters (e.g., ATP7A, ATP7B, MRP2) [55,56,57]. However, the database contains more than 40,000 species; many of them could probably be artifacts or contaminations. As many of these proteins were characterized in single studies only, their exosome localization should still be carefully confirmed.

Proteins that are commonly found in exosomes were designated as “vesicle-specific” markers. One of these protein families are tetraspanins, which are involved in the production of exosomes. This family includes CD9, CD63, and CD81 membrane proteins [58]. Tetraspanins have been suggested as biomarker candidates for various cancer and infectious diseases [59]. CD63^+^ exosomes were shown to be significantly increased in patients with melanoma [60] and other cancers [61], therefore CD63 has been suggested to be a “cancer biomarker”. Another member of the tetraspanin family, CD81, which plays an important role in cell entry of hepatitis C virus [62], has been demonstrated to be significantly upregulated on exosomes from serum of chronic hepatitis C patients [62]. It indicates that CD81 may be a marker for the diagnosis of HCV infection. Nevertheless, so far no universal exosome markers exist, as these groups of proteins are not always present in all exosomes and their quantity may vary dependent on the variable environmental conditions. Tetraspanins represent proteins frequently found in exosomes rather than universal (vesicle-specific) markers of exosomes. A search for a specific protein component of exosomes is still ongoing.

The full set of proteins present in exosomes is variable and reflects the phenotypic state of the cell from which they originate. For example, exosomes derived from T lymphocytes contain on their surface CD3 [63]. Content of exosomes can be modified in response to environmental changes. As an example, exosomes released from cells undergoing heat shock, show increased levels of heat shock proteins, such as HSP60, HSP70, or HSP90 [64].

## 5. Transition from In Vitro Preliminary Data into Clinical Research

A challenge in exosome research is connected with translating results from studies conducted in in vitro systems into in vivo studies. Most of the initial exosome studies utilized exosomes produced by cell lines and obtained from supernatants of these cells. Translating results from an in vitro experimental system, in which exosomes derive from a homogeneous cell population to an in vivo system with a heterogeneous exosome population released from healthy cells, immune cells, and tumor cells (TEX) is complicated due to difficulties in interpreting the results. Therefore, many study conditions need to be adapted.

Tumor-derived exosomes can interfere with the regulation of immune responses in the microenvironment of a tumor. They can affect the maturation of dendritic cells, impair the functioning of NK cells, induce suppressor cells originating from myeloma, and transform a macrophage phenotype into a pro-cancer one [1]. One of the basic functions assigned to exosomes released from tumor cells is the suppression of immune responses [65]. Exosomes contain immunoactive molecules (e.g., FasL, MICA/B, MHC, immunosuppressive cytokines, adhesion molecules, and enzymes, such as CD39 and CD73), which can modulate functions of immune cells [66]. It is also important that TEX carry tumor-associated antigens (TAA) and have the ability of their cross-presentation. Profiles of cancer cells and their TEX have similar expression patterns and comparable levels of TAA [67]. Once released to body fluids, exosomes produced by cancer cells containing tumor-associated antigens indicate the presence of a tumor. Utilization of this feature of exosomes enabled characterization of proteomes of pancreatic cancer cells and detection of enrichment in both pancreatic cancer cells and exosomes released by them in GPC1 proteoglycan (glypican 1). GPC1^+^ exosomes (crExos) were monitored in patient serum with the use of fluorescence-activated cell sorting (FACS) technique. It was demonstrated that the utilization of GPC1^+^ exosomes led to the successful detection of early stages of pancreatic cancer [68] and to the differentiation between patients with different stages of progression of this cancer.

Studies analyzing patient serum are still extremely rare. One reason could be the fact that there are still no markers allowing the safe identification of cancer-derived exosomes (TEX), except from melanoma [69,70]. Nevertheless, exosomes from serum or other biological fluids have an important potential as cancer biomarkers. In addition, they can possibly be used in cancer therapy as potent cancer vaccines or drug delivery systems. Targeting TEX in patients or inhibiting their release could provide an important therapeutic approach.

The analysis of exosomes that are present in the plasma of head and neck cancer (HNC) patients revealed that the profiles of immunomodulatory proteins and the ability to regulate functions of lymphocytes depend on the disease activity of the HNC patients [66]. One of the major, increasing risk factors in the pathogenesis of HNC is, besides alcohol and tobacco consumption, the human papillomavirus (HPV) infection [71,72]. The complete proteome of tumor-derived exosomes from HPV-positive and HPV-negative HNC cell lines was analyzed. Membrane components with putative immune-regulatory functions were evaluated and several proteins discriminating HPV-positive and HPV-negative cells were found. The observed specificity of proteins in exosomes with different immunomodulatory features could contribute to the different overall response of HPV-positive and HPV-negative cancers to the treatment [73].

Recent studies on novel immune checkpoint inhibitors, i.e., against CTLA-4 and PD-1 for different types of cancers, such as melanoma, lung cancer, and HNC led to new therapeutic perspectives for recurrent and metastatic tumor patients. The blockade of these receptors by anti-PD1 and anti-CTLA-4 inhibit the suppression of T cell activation and therefore restores T cell activity [74,75]. However, initial clinical studies revealed response rates in HNC of about 10–15% [76,77,78]. Hence, there is a need to identify determinants that are responsible for the resistance of these tumors to immunotargeted therapies. Lately, Poggio et al. proposed that PD-L1 exosomes promote tumor growth and their suppression restores anti-tumor immunity, which could be one major cause for resistance to immunotherapy [79].

## 6. Future Perspectives of Exosome Proteomics in Cancer Research

Proteomics of exosomes has undeniably made considerable progress in recent years. The improvement of exosome isolation methods was an essential factor that was responsible for this phenomenon, but also the development of the advanced instrumentation used for proteomic analysis and its elevated sensitivity allowed a significant improvement of the analysis. Nevertheless, there are still many missing points in exosome research. As an example, there are still no universal exosome markers, allowing clear identification of these vesicles and distinguishing them from other EVs. Moreover, standard methods of exosome characterization that are similarly working in vitro and in vivo are still not available.

There are several lines of evidence that, in cancer situations, exosomes may be key players involved in intercellular communication between tumor and non-tumor cells [1,2]. Exosomes potently contribute to many processes of tumorigenesis, such as trafficking of immune cells, generating the metastatic niche and modulation of tumor immune responses [80,81,82]. In metastatic melanoma, exosomes released by cancer cells influence the mobilization and recruitment of bone marrow cells to pre-metastatic niches, thus promoting metastasis [70]. For different cancers, distinct exosome-mediated mechanisms have been identified for dysregulated anti-tumor immune responses in patients [83,84]. One major cause of this could be the fact that, owing to their cargo, exosomes present in cancer patient plasma induce phenotypic alterations of immune cells present in the tumor microenvironment. Tumor-derived exosomes are considered to be responsible for delivering suppressive signals to immune cells, to induce apoptosis of cytotoxic T-cells, and to inhibit proliferation of natural killer (NK) cells [85]. Altogether, exosomes interfere with anti-tumor immunity and thus have an impact on patient response to the therapy. Since many exosomal proteins are associated with cell signaling, their role in the regulation of immune responses could be higher than initially anticipated. Therefore, the exploration of the immunomodulatory role of exosomes during cancer progression and the role of their protein cargo in this process (immunoproteomics) would be an important research direction in the future.

In summary, in the past decades, we have learned a great deal regarding a myriad of cargo molecules contained within exosomes and the complex roles exosomes play in the communication between tumor cells and their microenvironment. The accelerated research in the field of exosome research in the past five years has no doubt made great strides in understanding the complexities underlying the role of exosomes in cancer. However, due to the huge technical discrepancy in existing methods used to isolate, analyze, and characterize exosomes, the field of exosome research is still in its infancy. In order to use exosomes for diagnostic or prognostic monitoring of cancer patients, as well as to design novel exosome-based cancer therapies, these technical challenges have to be overcome.

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
