# Peer review of "Challenges in the Isolation and Proteomic Analysis of Cancer Exosomes—Implications for Translational Research"

_proteomes, 2019, doi:10.3390/proteomes7020022_

Reviewer 1 Report

As someone interested in the development of new approaches to isolate exosomes, I was excited by the prospect of a review that focuses specifically on the challenges of sample prep and its implications for proteomic studies.  The title and abstract make it abundantly clear that this was the intent of this review.  Unfortunately, the manuscript does very little to address this primary topic.  And the bulk of what is presented essentially repeats the standard beats describing simple background, with very little insight.

Line 126-131 represent the only discussion relevant to methods of isolation (with 3 references).  The authors seem to suggest that affinity chromatography is superior (line 135 - AC has a clear advantage), but offer no references to support this.  In the end, the true scope of this review (exosome isolation) is summarized by pointing to another review (Xu, 2016) on this very topic

The discussion of sources of exosomes (cell line, blood, urine, CF) is presented with little to no insight, and does not inform the reader of relevant strategies to pursue their goals.  As an example,  (line 102)  states that "Blood contains many molecules" and that (line 103) "this is one factor which should be eliminated in order to obtain pure exosomal fraction".  These statements are not only exceptionally obvious, but reinforce this reviewers opinion that the manuscript offers no tangible benefits to the research community.  There are countless statements such as this (line 155 "Concluding, urine seems to be a good source...") which point this review as being superficial at best.

Similar comments can be made about the proteomic discussion.  Statements are made without approprraite references (eg line 156-158).

In other cases, statements appear oversimplified, or misquote the source (line 60 quotes ref 8 which states that "most exomes have an evolutionarily conserved set of proteins, including tetraspanins, Alix, and Tsg101..."  This does not imply that the entire exosome protein cargo is conserved in all exosomes, as line 60 seems to suggest.  Also, ref 8 seems to point that there are several preferred protein markers of exosomes, while the authors labor on the point that universal markers do not exist.  But is this not the very intent of all the reports that demonstrate specific biomarkers to specific conditions (eg tumours).

The references do not accurately represent the state of the art in exosome research, and are perhaps biased towards specific authors.  Some references appear to have been discredited by other works. 

The future perspectives section offers little to no real insight.  And there are too many instances of grammatical errors or confusing sentences to point out here.

It is suggested that the authors attempt a more focused review, rather than a very broad, and overly simplified overview which does not present any new insight. 

I cannot recommend publication of this manuscript

Author Response

Response to the Reviewer 1.

As someone interested in the development of new approaches to isolate exosomes, I was excited by the prospect of a review that focuses specifically on the challenges of sample prep and its implications for proteomic studies.  The title and abstract make it abundantly clear that this was the intent of this review.  Unfortunately, the manuscript does very little to address this primary topic.  And the bulk of what is presented essentially repeats the standard beats describing simple background, with very little insight.

We agree with the reviewer and in the revised version, we have described the current state of art and open questions in the field. Please see below point-by-point response to the reviewers comment.

Line 126-131 represent the only discussion relevant to methods of isolation (with 3 references).  The authors seem to suggest that affinity chromatography is superior (line 135 - AC has a clear advantage), but offer no references to support this.  In the end, the true scope of this review (exosome isolation) is summarized by pointing to another review (Xu, 2016) on this very topic

To address this point we have now included references for each exosome isolation methods including affinity chromatography. For more insight we have also included a paragraph (231-234) mentioning the current state of art to combine more than one isolation techniques towards achieving pure population of exosomes for molecular analysis, like proteomics.

The discussion of sources of exosomes (cell line, blood, urine, CF) is presented with little to no insight, and does not inform the reader of relevant strategies to pursue their goals.  As an example,  (line 102)  states that "Blood contains many molecules" and that (line 103) "this is one factor which should be eliminated in order to obtain pure exosomal fraction".  These statements are not only exceptionally obvious, but reinforce this reviewers opinion that the manuscript offers no tangible benefits to the research community.  There are countless statements such as this (line 155 "Concluding, urine seems to be a good source...") which point this review as being superficial at best.

We agree with the reviewer comment and have now provided more insight by providing examples on the proteins or contaminants co-isolated during exosome isolation from cell culture supernatant or bodily fluid (blood, urine, and CNS). Moreover, we have provided insight into the approaches and challenges faced to deplete these contaminating signal from exosome meant for proteomic analysis. The modified text has been included in section Source and methods of isolation for proteomics analysis.

Similar comments can be made about the proteomic discussion.  Statements are made without approprraite references (eg line 156-158).

Thank you for this comment. We corrected this accordingly

In other cases, statements appear oversimplified, or misquote the source (line 60 quotes ref 8 which states that "most exomes have an evolutionarily conserved set of proteins, including tetraspanins, Alix, and Tsg101..."  This does not imply that the entire exosome protein cargo is conserved in all exosomes, as line 60 seems to suggest.  Also, ref 8 seems to point that there are several preferred protein markers of exosomes, while the authors labor on the point that universal markers do not exist.  But is this not the very intent of all the reports that demonstrate specific biomarkers to specific conditions (eg tumours).

We thank the reviewer for the critical insight, we have now taken care of this comment in the revised version and deleted oversimplified statements and misquoted sources. Please see the modified text of section Exosomes and their function.

The references do not accurately represent the state of the art in exosome research, and are perhaps biased towards specific authors.  Some references appear to have been discredited by other works. 

In the revised version, we have tried our best to include key references in the field.

The future perspectives section offers little to no real insight.  And there are too many instances of grammatical errors or confusing sentences to point out here.

As suggested by the reviewer, we have expanded the future perspective section and have  corrected grammatical errors accordingly

It is suggested that the authors attempt a more focused review, rather than a very broad, and overly simplified overview which does not present any new insight. 

To improve our manuscript, we have now revised it accordingly to the reviewer suggestions. We feel that our review in the current form presents focused and up-to-date view on the research field of exosome proteomics.

Reviewer 2 Report

Dear authors,

The review “Challenges in the isolation and proteomic analysis of cancer exosomes – implications for translational research” offers a good overview of the possibility to use EXOs and their proteomic for downstream clinical applications; while not going into details, it touches upon all the most relevant aspects for the topic under discussion.

This reviewer requires only few minor corrections.

Line 51: small membrane vesicles … membrane-bound structures… Please, rephrase the sentence, it seems a redundancy

Line 52-53: To give a more complete view of the biogenesis of exosomes it would be advisable to add a brief mention to the conclusive part of the process, i.e. the fusion of the MVB with the membrane and the release of the EXOs

Line 63: The cited paper (Zhang H et al. Identification of distinct nanoparticles and subsets of extracellular vesicles by asymmetric flow field-flow fractionation. Nat Cell Biol. 2018 Mar;20(3):332-343. doi: 10.1038/s41556-018-0040-4.) defines exomeres as “non-membranous nanoparticles” (“we identified two exosome subpopulations (large exosome vesicles, Exo-L, 90-120 nm; small exosome vesicles, Exo-S, 60-80 nm) and discovered an abundant population of non-membranous nanoparticles termed 'exomeres' (~35 nm)”). The authors referred the term exomeres to Exo-S, 60-80 nm. Please correct.

Line 75: “exosomes are involved in many aspects of intracellular…” maybe authors mean intercellular. Please check.

Line 129: squared brackets contain reference 23 twice. Please check.

Line 163-164: the authors affirm that “the largest attention is paid to exosomes because of their role in cell-to—cell communication”... MVs, as well as EXOs, are involved in cell-to-cell communication and may mediate similar or overlapping functions. Thus, it would be advisable to reformulate the sentence so that it is more scientifically correct.

Line 232: please explain what HNC stands for.

Author Response

Response to Reviewer 2 Comments

The review “Challenges in the isolation and proteomic analysis of cancer exosomes – implications for translational research” offers a good overview of the possibility to use EXOs and their proteomic for downstream clinical applications; while not going into details, it touches upon all the most relevant aspects for the topic under discussion.

A few minor corrections:

Line 51: small membrane vesicles … membrane-bound structures… Please, rephrase the sentence, it seems a redundancy

We agree with the Reviewer and corrected it accordingly

Line 52-53: To give a more complete view of the biogenesis of exosomes it would be advisable to add a brief mention to the conclusive part of the process, i.e. the fusion of the MVB with the membrane and the release of the EXOs

The reviewer is right - we implemented this information accordingly (line 66-67)

Line 63: The cited paper (Zhang H et al. Identification of distinct nanoparticles and subsets of extracellular vesicles by asymmetric flow field-flow fractionation. Nat Cell Biol. 2018 Mar;20(3):332-343. doi: 10.1038/s41556-018-0040-4.) defines exomeres as “non-membranous nanoparticles” (“we identified two exosome subpopulations (large exosome vesicles, Exo-L, 90-120 nm; small exosome vesicles, Exo-S, 60-80 nm) and discovered an abundant population of non-membranous nanoparticles termed 'exomeres' (~35 nm)”). The authors referred the term exomeres to Exo-S, 60-80 nm. Please correct.

We corrected it accordingly – see lines: 99-109

Line 75: “exosomes are involved in many aspects of intracellular…” maybe authors mean intercellular. Please check.

The reviewer is right; it should indeed be “intracellular”. We corrected it accordingly (line 87)

Line 129: squared brackets contain reference 23 twice. Please check.

We corrected it accordingly

Line 163-164: the authors affirm that “the largest attention is paid to exosomes because of their role in cell-to—cell communication”... MVs, as well as EXOs, are involved in cell-to-cell communication and may mediate similar or overlapping functions. Thus, it would be advisable to reformulate the sentence so that it is more scientifically correct.

Thank you for your comment. Due to its redundancy, we havedecided to removed this sentence from the manuscript

Line 232: please explain what HNC stands for.

We agree with the reviewer and corrected it accordingly

 Reviewer 3 Report

The authors systematically discussed exosomes and their function, source and methods of isolation of exosomes for proteomics study, the advantages and disadvantages of current extraction and analysis methods, and finally gave examples in transition to clinical research and future perspectives. It is a very inspiring topic to review and will help researchers to study cancer exosome based on clinical needs. Meanwhile, the current limits and challenges were listed clearly in the article, which would help readers avoid the most common mistakes that may be made. It is a good article to be accepted by Proteomes after minor revision.

Comments:

1.     There is a recently published paper “Suppression of Exosomal PD-L1 Induces Systemic Anti-tumor Immunity and Memory” in Cell by Poggie and Blelloch, etc. (April 4th, 2019), which is very impressive research talking about the communication function of cancer exosomes carrying PD-L1, in not only clinical diagnostics but also oncotherapy. It will make the article more profound if the authors can include discussion of the exosomal PD-L1, which is the key research areas of macromolecular drugs in oncotherapy.

2.     In page 232, What is HNC disease? One should label it clearly since it is the first time described in the article. Same as in Page 235, what is HPV positive/negative statuses? It is better to rewrite the whole paragraph to make it easy for readers to understand.

3.     There is a “1” in front of “Introduction”, which is redundant.

4.     Page 62, “... subjected to the asymmetric flow field flow fractionation ...” is containing grammatical mistakes. Please change all similar mistakes throughout the article.

Author Response

Response to Reviewer 3 comments:

The authors systematically discussed exosomes and their function, source and methods of isolation of exosomes for proteomics study, the advantages and disadvantages of current extraction and analysis methods, and finally gave examples in transition to clinical research and future perspectives. It is a very inspiring topic to review and will help researchers to study cancer exosome based on clinical needs. Meanwhile, the current limits and challenges were listed clearly in the article, which would help readers avoid the most common mistakes that may be made. It is a good article to be accepted by Proteomes after minor revision.

Comments:

1.     There is a recently published paper “Suppression of Exosomal PD-L1 Induces Systemic Anti-tumor Immunity and Memory” in Cell by Poggie and Blelloch, etc. (April 4th, 2019), which is very impressive research talking about the communication function of cancer exosomes carrying PD-L1, in not only clinical diagnostics but also oncotherapy. It will make the article more profound if the authors can include discussion of the exosomal PD-L1, which is the key research areas of macromolecular drugs in oncotherapy.

Thank you for this important comment. We were not aware of this manuscript. Since it is highly important and relevant topic, we included this into our manuscript (see Line 342-350 in the reviewed version with marked changes)

2.     In page 232, What is HNC disease? One should label it clearly since it is the first time described in the article. Same as in Page 235, what is HPV positive/negative statuses? It is better to rewrite the whole paragraph to make it easy for readers to understand.

The reviewer is right, we introduced HNC and HPV terms and rewrote the paragraph to provide a better clarity

3.     There is a “1” in front of “Introduction”, which is redundant.

We apologize for this mistake, we overlooked it. We corrected it accordingly

4.     Page 62, “... subjected to the asymmetric flow field flow fractionation ...” is containing grammatical mistakes. Please change all similar mistakes throughout the article.

We have read and corrected grammatical mistakes throughout the article.